# A Novel Self-Assembled Graphene-Based Flame Retardant: Synthesis and Flame Retardant Performance in PLA

**DOI:** 10.3390/polym13234216

**Published:** 2021-12-01

**Authors:** Peixin Yang, Hanguang Wu, Feifei Yang, Jie Yang, Rui Wang, Zhiguo Zhu

**Affiliations:** School of Materials Design and Engineering, Beijing Institute of Fashion Technology, Beijing 100029, China; 17862989137@163.com (P.Y.); yangfei20211120@163.com (F.Y.); a1051154816@163.com (J.Y.); clywangrui@bift.edu.cn (R.W.)

**Keywords:** flame retardant, graphene, melamine, phytic acid, nucleation effect

## Abstract

In this study, a novel flame retardant (PMrG) was developed by self-assembling melamine and phytic acid (PA) onto rGO, and then applying it to the improvement of the flame resistance of PLA. PMrG simultaneously decreases the peak heat release rate (pHRR) and the total heat release (THR) of the composite during combustion, and enhances the LOI value and the time to ignition (TTI), thus significantly improving the flame retardancy of the composite. The flame retardant mechanism of the PMrG is also investigated. On one hand, the dehydration of PA and the decomposition of melamine in PMrG generate non-flammable volatiles, such as H_2_O and NH_3_, which dilute the oxygen concentration around the combustion front of the composite. On the other hand, the rGO, melamine, and PA components in PMrG create a synergistic effect in promoting the formation of a compact char layer during the combustion, which plays a barrier role and effectively suppresses the release of heat and smoke. In addition, the PMrGs in PLA exert a positive effect on the crystallization of the PLA matrix, thus playing the role of nucleation agent.

## 1. Introduction

As an excellent bio-degradable material with extraordinary features, such as its good mechanical properties, good processing ability, and biological compatibility, polylactic acid (PLA) has attracted increasing attention due to its wide application in fields including biomedical engineering, textiles, and electronic equipment [1,2]. However, PLA presents poor flame retardancy, and its limiting oxygen index (LOI) value is only 19. The poor flame retardancy of PLA significantly limits its application, especially in some military fields. PLA products with improved fire resistance properties are highly desired in fields including the electronic, automotive, and aerospace industries [3,4,5]. A variety of strategies have been used to improve the flame retardancy of PLA, including flame retardant finishing and the addition of flame retardants (e.g., phosphorus-based, nitrogen-based, silicon-based flame retardants) [6,7,8]. Phytic acid (PA) was developed as a kind of phosphorus-based flame retardant for PLA, which shows great potential due to its non-toxicity [9]. Tang et al. used PA aqueous solution to treat PLA non-woven fabrics to endow the fabric with excellent flame retardancy even after multiple washes [10]. However, PA induces the decomposition of the composite when it is used alone as the flame retardant, thus shortening the ignition time. Therefore, in order to achieve a better flame retardant performance, PA is commonly used after modification or in company with other materials to prepare flame retardants. Zhang et al. fabricated a flame retardant with encapsulated microstructure, in which the PA inserted layered double hydroxide functioning as the core structure, and the mixture of PA and casein was used as the shell material [11,12]. They applied the microcapsule in the fire resistance of PLA, which increased the LOI value from 19.6% to 28.3%. Additionally, the vertical burning (UL-94) rating of the obtained composite was upgraded from no rating to V-0, and the peak heat release rate (pHRR) value was decreased from 935.8 kW/m^2^ to 747.9 kW/m^2^, indicating the excellent flame resistance effect of this PA-based flame retardant on PLA. In Hu’s work, the authors synthesized a flame retardant for PLA by reacting PA, calcium chloride, and magnesium chloride, which significantly reduced the pHRR of the modified PLA by 33% [13,14]. Zhou prepared a bio-based flame retardant for PLA by using PA and chitosan, and found that 3 wt% PA/chitosan exhibited a clear enhancement effect on the LOI value of PLA from 19.6% to 30.5%, and the PLA/PA/chitosan composite achieved a V-0 grade on a UL-94 test with slight melt dripping [15]. In Yang’s work, a novel bio-based flame retardant (PA-THAM) was prepared through a salt formation reaction between PA and trometamol, which significantly reduced the molten viscosity of the composite when added to PLA [16].

Melamine is a typical component in the intumescent flame retardant system, which is usually used cooperatively with the phosphorus-based flame retardants to further improve its fire resistance effect [17,18,19]. Sun et al. incorporated melamine polyphosphate (MPP) and zinc bisdiethylphosphinate (ZnPi) into PLA, and they found that the PLA composite with 15 wt% of MPP/ZnPi (3:2) exhibited the best flame-retardant efficiency, with a LOI value of 30.1% and a V-0 grade on UL-94 tests [20]. In Wang’s work, melamine was used in combination with PA during the fabrication of the flame retardant for epoxy resin, which decreased the pHRR, total smoke release (TSR), and fire growth rate of the epoxy resin-based composite by 62.3%, 36.2%, and 62.16%, respectively [21]. However, the poor compatibility of melamine with the majority of polymer materials deteriorates its dispersion in composites, creating a negative effect on its flame resistance [22,23]. Therefore, researchers try to add some carbon nano-layered materials, such as graphene, carbon nanotubes, and others. As a nonflammable material with a multilayer structure, graphene offers good heat insulation and higher thermal stability; thus, it has also been used as a component of various flame-retardant systems [24,25,26,27]. In Song’s work, the authors prepared supramolecular aggregates as the flame-retardant for epoxy resin through self-assembling piperazine and PA onto reduced graphene oxide (rGO), and the tri-component cooperative effect effectively improved the flame-retardant performance of the epoxy resin-based composite [28]. Yuan et al. modified graphene oxide (GO) by using melamine to endow the GO with good dispersibility and high flame retardant efficiency, and added the obtained functional GO/melamine into PP to improve the thermal stability and flame retardancy of the composite [29].

In this study, we developed a novel flame retardant for PLA through the self-assembly of rGO, melamine, and PA, which is abbreviated as PMrG. The morphology and chemical structure of PMrG was studied. Next, the effects of PMrG on the thermal stability and flame retardancy of PLA were systematically investigated; the mechanism was also discussed in this article. By integrating the phosphorus supplying effect of the PA component, the intumescent effect of the melamine component, and the flame retardant skeleton constructed by the rGO nanolayers, PMrG exhibited a remarkable flame retardant effect on PLA. PMrG exerted the best flame retardant effect on PLA when the ratio of rGO, melamine, and PA was controlled at 1:1:5, which provided the composite with the highest LOI value, the lowest heat release rate (HRR), total heat release (THR), and total smoke release (TSR) values, and the longest time to ignition (TTI) when the amount of the additive is maintained at 10 wt%. By using this flame retardant, the composite achieved a V-0 grade on the UL-94 test. In addition, the pHRR of the composite decreased by 35%, and the total heat release decreased by 21%. This study provides a simple and green approach for creating highly effective graphene-based flame retardants for PLA.

## 2. Experimental Section

### 2.1. Materials

The reduced graphite oxide (rGO) was prepared by using Hummers’ method [30]. The PA (50 wt%) was provided by Tokyo Chemical Industry Co. Ltd. (Tokyo, Japan), the melamine was purchased from Shanghai Macklin Biochemical Co. Ltd., and the PLA was supplied by Nature Works Company (Blair, NE, USA). The dichloromethane was supplied by Beijing Chemical Works Co. Ltd (Beijing, China).

### 2.2. Synthesis of PMrG

A total of 0.2 g Melamine and 0.9 mL PA were successively added into 100 mL rGO aqueous suspension (1.8 mg mL^−1^). After a 12 h reaction under mild stirring at room temperature, the flame retardant system composed of rGO, melamine, and PA was obtained, which is named PMrG in this article. By controlling the ratio of precursors, the PMrG samples with different amounts of melamine and PA were synthesized for comparation, and the mass ratio of melamine and PA remained constant at 1:5 (as shown in Appendix A).

In order to illustrate the synergistic effect of rGO, melamine, and PA in PMrG on its flame resistance, we also fabricated a comparative subject by gently stirring 0.2 g melamine and 0.9 mL PA at 100 mL aqueous solution, which is abbreviated as PM.

### 2.3. Preparation of PMrG/PLA Composites

After being washed by using ethanol, 0.6 g prepared PMrG was dispersed in 80 mL dichloromethane to produce the PMrG/dichloromethane suspension. Next, 5.4 g PLA was added into the PMrG/dichloromethane suspension under mild stirring to dissolve the PLA homogeneously, and the PMrG/PLA/dichloromethane mixture was obtained, in which the mass ratio of PMrG and PLA was controlled at 1:9. Subsequently, the PMrG/PLA/dichloromethane mixture was poured into the Teflon mould and placed at 60 °C under vacuum to remove the dichloromethane solvent until no bubbles emerged, and the PMrG/PLA composite with 10 wt% PMrG was prepared.

The rGO/PLA composite with 10 wt% rGO and PM/PLA composite with 10 wt% PM were also prepared through dichloromethane dissolution for comparative study.

### 2.4. Characterization

Fourier transform infrared spectrometry (FTIR) was conducted by a Nicolet 6700 FTIR spectrometer (Thermo Fisher Scientific, Waltham, MA, USA) with an attenuated total reflection (ATR) accessory over the range of 400–4000 cm^−^^1^. The X-ray photoelectron spectroscopy (XPS) was performed on an X-ray photoelectron spectrometer (Thermo ESCALAB 250, Waltham, MA, USA); the emission current and voltage were controlled at 5 mA and 10 kV, respectively. Raman spectroscopy was conducted by a SPEX-1403 laser Raman spectrometer (Renishaw inVia, London, UK) at an excitation wavelength of 532 nm. The morphologies of the samples were observed by using a JSM-7500F scanning electron microscope (SEM, JEOL, Tokyo, Japan). The mechanical performances were measured by using an Instron electronic universal material strength tester (4302, Instron Corporation, High Wycombe, UK) at a speed of 10 mm·min^−^^1^ according to the ASTM D-638 standard. The width of the specimen was 4.0 ± 0.1 mm and the thickness was 2.0 ± 0.1 mm. All the samples were tested five times and the average values were selected as the final results.

Thermogravimetric analysis (TGA) and differential scanning calorimetry (DSC) were used to characterize the thermal behavior of the samples. The TGA was conducted by using a TG 209 (Netzsch, Germany), and the samples were heated from 30 °C to 700 °C at a rate of 10 °C min^−^^1^. The DSC was conducted by using a Q2000 (TA instruments, USA). The samples were first heated from 30 °C to 200 °C to eliminate their thermal history, and then they were cooled from 200 °C to 30 °C at a rate of 10 °C min^−^^1^.

The flame retardant properties of the neat PLA and the PLA-based composites were investigated by using LOI, vertical burning testing (UL-94), and cone calorimeter tests (CCTs). The LOI tests were conducted on a Dynisco LOI test instrument according to the ASTM D 2863-97 standard. The size of the specimen was 80 mm *×* 6.5 mm *×* 3 mm. The UL-94 tests were carried out on a vertical burning instrument (CFZ-2, Jiangning Analytical Instrument Factory Co., Ltd., Nanjing, China), according to the ASTM D3801 standard. The size of the specimen was 130 mm × 13 mm × 3 mm. The CCTs were conducted by using a calorimeter (iCone, Fire Testing Technology Co., Ltd., UK) according to the ISO 5660-1 standard under a heat flux of 35 kW m^−^^2^, and the sample with the dimensions of 100 mm × 100 mm × 3 mm was wrapped with aluminum foil.

## 3. Results and Discussion

### 3.1. Fabrication of PMrG

PMrG was prepared as the precipitate after a simple self-assembling of rGO, melamine, and PA in the aqueous solution, which is demonstrated in Figure 1a. The obtained PMrG presented as clusters with a hierarchical structure, in which melamine and PA aggregate on the rGO multilayer skeleton (see Figure 1b). By controlling the ratio of precursors, the PMrG samples with different amounts of melamine and PA attachments on rGO were synthesized, and the ratio of melamine and PA remained constant (see Appendix A). In order to optimize the flame retardancy of the composite, the PMrG with the rGO/melamine/PA ratio of 1:1:5 (PMrG-3) was selected as the additive of PLA for testing.

In the infrared spectrum (FTIR) of the collected PMrG, the absorption peaks at 778 cm^−1^, 1510 cm^−1^, 1668 cm^−1^ (C=N), and 3358 cm^−1^ (-NH_2_) are assigned to the triazine rings from melamine (see Figure 1c) [31]. Compared with the FTIR spectra of rGO and melamine, the FTIR spectrum of PMrG demonstrated peaks at 924 cm^−1^ (P-OH), 1050 cm^−1^ (P-O-C), and 1135 cm^−1^ (P=O), which are assigned to PO_4_^3−^ groups resulting from the PA, indicating the successful combination of rGO, melamine, and PA in the PMrG. In the X-ray photoelectron spectrum (XPS) of the PMrG (Figure 1d and Appendix A), the peaks at 133.2 eV and 134.1 eV are related to P=O and P-O-C, and the peaks at 285 eV, 400 eV, and 534.1 eV are associated with C atoms, N atoms, and O atoms, respectively. In the high-resolution XPS spectrum for C 1s (Appendix A), the main peak centered at 284.6 eV is attributed to the graphitic sp2 carbon, whereas the additional component centered at 285.8 eV is assigned to C-N and or C=N. The peaks in the XPS spectrum for N 1s (Appendix A) are attributed to pyridinic (398.5 eV) and pyrrolic (400.1 eV), respectively. In addition, the XPS spectrum for P2p (Appendix A) deconvolutes into two peaks at 133.2 eV and 134.1 eV, corresponding with P=O and P-O-C, respectively. The XPS results indicate the formation of the N, P-doped carbon in PMrG, which is consistent with the FTIR results. The thermal degradation behaviors of the rGO, PM, and PMrG are shown by their TGA curves (see Figure 1e and Appendix A), in which the temperature at 5% weight loss occurs (T_5%_), the temperature of maximum weight loss rate (T_max_), and the residual weight at 700 °C (RW) can be obtained. Compared with rGO, PMrG-3 degrades at a lower temperature due to the poor thermal stability of PA within it. However, the degradation temperature of PMrG-3 is much higher than that of PM, which should be attributed to the more ordered structure of melamine and PA in PMrG provided by the rGO skeleton. In addition, the RW of PMrG (45 wt%) is much higher than that of rGO (38.4 wt%), indicating the promotion effect of rGO, melamine, and PA components in PMrG on char formation during thermal degradation.

### 3.2. Fabrication of PMrG/PLA Composite

The synthesized PMrG was used as a flame retardant in the PLA, and the fabrication process of the PMrG/PLA composite is shown in Figure 2a. In order to achieve a good dispersion state, we prepared the PMrG/dichloromethane suspension and then added PLA powders into the suspension to achieve a complete dissolution. The mass ratio of PMrG to PLA was controlled at 1:9. After the removal of the dichloromethane solvent, the PMrG/PLA composite with 10 wt% PMrG was prepared (see Figure 2b). Due to the good dispersion of the PMrG and the integrity of the internal structure (see Figure 1c), the obtained PMrG/PLA composite demonstrated a good mechanical performance (see Figure 1d).

The crystallization and thermal properties of the PLA and its composites were studied by using DSC, and the obtained curves are shown in Figure 3. It can be seen that the DSC curves of the PLA composites during the cooling process demonstrated a clear crystallization peak (Figure 3a), indicating that rGO, PM, and PMrG-3 all act as nucleation agents in PLA composites, which increases the crystallinity of PLA. Compared with the neat PLA and the rGO/PLA composite, the PM/PLA and PMrG-3/PLA composites exhibited much clearer crystallization peaks at higher temperatures. In addition, the crystallization temperature of the PMrG/PLA composite increased significantly with the increased amount of melamine and PA components (see Appendix A). Therefore, the nucleation promotion effect of PMrG was mainly due to the melamine and PA components. Furthermore, the disappeared cold crystallization peaks of the PM/PLA and PMrG-3/PLA composites also demonstrate their improved crystalline ability (Figure 3b), while the glass transition temperature and melting temperature of PLA were barely influenced.

In order to further study the influence of rGO, PM, and PMrG on the thermal properties of PLA, TGA was carried out in an N_2_ atmosphere to assess the thermal stability and char-forming ability of the PLA and the composites. The TGA and derivative thermogravimetry (DTG) curves of neat PLA and its composites are shown in Figure 4 and Appendix A, and the related thermal data are summarized in Appendix A. The TGA curve of neat PLA presents the one-step thermal degradation process at 357.6 °C, with a maximum weight loss rate, and the char residue is almost 0%. The incorporation of rGO, PM, and PMrG into the PLA matrix enhanced the degradation temperature of the composites. When 10 wt% rGO was added to PLA, T_5%_ increased from 321.1 °C to 338.6 °C, and T_max_ increased to 367.5 °C, which was mainly attributed to the good barrier effect and heat transfer insulation of the graphene layers [32]. When the PLA was incorporated with 10 wt% PM, the T_max_ increased to 368.4 °C. In addition, the composites with PLA and 10 wt% PMrG also exhibited increased T_5%_ and T_max_ values compared with the neat PLA, demonstrating the improved thermal stability and delayed decomposition of PLA after the addition of PMrG. According to the results of the DSC test (Figure 3), we attributed the improved thermal stability of the rGO/PLA, PM/PLA, and PMrG/PLA composites to the improved crystalline ability of PLA under the function of the additives. When the composition of PMrG changed, the degradation temperature of the composite was hardly influenced (see Appendix A), while the peak value of DTG was significantly affected due to the significant change in the char residue (see Appendix A). The char residue reached its peak value (6.3%) when the ratio of rGO, melamine, and PA in the PMrG was controlled at 1:1:5, which is defined as PMrG-3. The RWs of the rGO/PLA, PM/PLA, and PMrG-3/PLA were 7.8%, 4.2%, and 6.3%, respectively, which were much higher than that of the neat PLA, indicating the significant improvement in the residual yield of the composites.

### 3.3. Flame-Resistant Effect of PMrG on PLA

The flame resistance properties of the neat PLA and the PLA-based composites were investigated by using LOI, vertical burning testing (UL-94), and cone calorimeter tests (CCT). The LOT value is commonly regarded as the most important specific criteria for the flame resistance of materials, and the results of the LOI tests in our study are shown in Figure 5 and Appendix A. For the neat PLA, the LOI value was only 19%, and it could not pass any grade on the UL-94 test. The LOI value of the rGO/PLA increased to 22%, and it was rated as V-1 grade. When 10% PM was added into the PLA, the LOI value increased to 23%, but exhibited no rate on the UL-94 test. In addition, the same content of PMrG-3 imparted a higher LOI value (25%) to the PLA than that of rGO or PM alone, indicating the higher flame-resistant effect of PMrG. The results obtained from the UL-94 test indicate that the addition of 10 wt% PMrG-3 can produce a V-0 rating, indicating that the combination of rGO and PM in PMrG shows an excellent synergistic effect on the flame resistance improvement of PLA.

CCT is widely used to evaluate the combustion performance of polymers under a forced-flaming fire scenario. In our study, the cone calorimetric parameters, including the TTI, HRR (including pHRR), THR, TSR, and char residue were recorded by using CCT to evaluate their fire and smoke risk, and the detailed data are collected in Table 1 and Figure 6.

The HRR and THR curves of the neat PLA and the PLA-based composites are shown in Figure 6a,b, respectively. It can be seen that the neat PLA burned at 74 s, showing a single pHRR of 426.6 kW/m^2^ at 170 s, and the THR reached 58.8 MJ/m^2^. The pHRR for the composite containing 10 wt% rGO reduced by 27.5% to 309.1 kW/m^2^, while the THR displayed no significant change compared with the neat PLA. In addition, the rGO/PLA composite ignited earlier than the neat PLA. When 10 wt% PM was added into the PLA, the obtained composite exhibited a sharp HRR peak with pHRR of 456.6 kW/m^2^. However, the THR significantly reduced, by 26.55%, to 43.2 MJ/m^2^ compared with that of the neat PLA. The incorporation of 10 wt% PMrG resulted in an earlier TTI and a decreased pHRR value, which is similar to the effect created by the addition of rGO. The pHRR values of the PMrG/PLA composites were even lower than those of rGO/PLA (Table 1 and Appendix A). Furthermore, the PMrG/PLA composite also exhibited a low THR value (Table 1 and Appendix A). Therefore, as the self-assembling product of rGO, melamine, and PA, PMrG endows PLA composites with a slower heat release rate and a reduced THR simultaneously. It can be seen from Appendix A that the PMrG/PLA composite with PMrG-3 as the flame retardant exhibited the latest TTI (59 s), the lowest pHRR (276.1 kW/m^2^), and lower THR (46.5 MJ/m^2^) values, indicating that PMrG exerts the best flame retardant effect on PLA when the ratio of rGO, melamine, and PA is 1:1:5.

The smoke release of materials is regarded as one of the major factors leading to death, and a lower smoke production rate and TSR denote a lower smoke risk and longer escape time in a fire disaster. The TSR of the neat PLA and the PLA-based composites are shown in Figure 6c, and the detailed data are listed in Table 1 and Appendix A. It can be seen that the neat PLA hardly released smoke during its burning process, but the addition of rGO or PM significantly increased the amount of smoke released. This was mainly attributed to the incomplete combustion of the composites caused by the rGO or PM. Therefore, when rGO or PM were added into PLA to decrease the heat release, the smoke suppression of the composites deteriorated at the same time. When 10 wt% PMrG was added as the flame retardant, the obtained PMrG/PLA composite presented a decreased TSR value with the increased amount of melamine and PA within the PMrG (see Appendix A), indicating that when the ratio of rGO, melamine, and PA is controlled at a specific range, the obtained PMrG product can overcome the smoke releasing problem brought by rGO and PM to some extent. The PMrG-3/PLA composite exhibited the best smoke suppression ability, and its TSR value was as low as 191.5 m^2^/m^2^ (see Figure 6c). From the char residue values of the PMrG/PLA composites (Appendix A), it can be seen that the PMrG-3/PLA composites left the most char residue (15.3%) after the combustion; thus, the low TSR value of the PMrG-3/PLA composites was partly attributed to its higher char residue. Therefore, the simultaneous presence of rGO, melamine, and PA provided the PMrG with a synergistic effect on the smoke suppression of the PMrG/PLA composite, and the effect was most significant when the ratio of rGO, melamine, and PA was controlled at 1:1:5.

Therefore, the PMrG/PLA composite with PMrG-3 as the flame retardant reached the V-0 grade, which simultaneously exhibited the highest LOI value, the longest TTI, and the lowest pHRR, THR, and TSR values, indicating the excellent flame-resistant effect and smoke suppression effect of PMrG-3 on PLA.

### 3.4. Mechanism for the Flame Retardant Effect of PMrG

In order to reveal the mechanism for the flame resistance of PMrG on PLA, the morphology and structure of the char residues after CCT of the neat PLA and the PLA composites were observed. The digital photos and SEM images are shown in Figure 7. It can be observed that the neat PLA formed almost no char residue (Figure 7a,e), leading to a weak barrier effect. The PLA composite with 10 wt% rGO formed more char than the neat PLA, but there were cracks and large cavities in the char residue structure (Figure 7b,f), limiting the barrier effect of the rGO on the burning of the rGO/PLA composite. The PM/PLA composite formed a swelling char layer after the combustion (Figure 7c,g). This should be attributed to the melamine component in PM, which functions as gas source in intumescent flame retardants and decreases the THR value. However, large cracks and cavities still formed in the char layer, resulting in an inferior heat isolation effect of the PM/PLA composite during combustion; thus, the PM/PLA composite still presented high HRR and TSR values. When 10 wt% PMrG-3 was added into the PLA as the flame retardant, the obtained PMrG-3/PLA composite formed a continuous swelling char with a much more compact structure after the combustion, and almost no crack or cavity was present in it. This dense char layer functioned as an intact shield, which effectively suppressed the transfer of heat and the supply of combustible gases during the combustion of the PMrG/PLA composite, leading to low THR and TSR values in the PMrG-3/PLA composite.

To further clarify the flame retardant mechanism of PMrG on PLA, Raman spectroscopies of the char residues after CCT were conducted. The obtained spectra are shown in Figure 8. The graphitization degree of the char residue is always reflected by the ratio of I_D_ and I_G_. A lower I_D_/I_G_ value means a higher graphitization degree, indicating a denser char layer structure [28]. For the neat PLA, the residual carbon was hardly formed after its combustion; thus, no continuous carbon layer was formed. The I_D_/I_G_ values of the rGO/PLA and PM/PLA composites were 1.22 and 1.2, respectively (see Figure 8a,b). By contrast, the PMrG-3/PLA composite exhibited an I_D_/I_G_ value of 1.08 (see Figure 8c), indicating that PMrG-3 clearly enhances the graphitization degree of char layers, which is strongly consistent with the SEM images in Figure 7.

Furthermore, the compositions of char residues were analyzed by XPS (see Figure 9). It can be seen that the XPS spectra of all the samples displayed sharp peaks at 285.8 eV and 533.5 eV, which corresponded to the C and O elements, respectively. For the PMrG-3/PLA composite, a P2s peak at 135.9 eV, a P2p peak at 190.5 eV, and an N1s peak at 399.4 eV can be observed in its XPS spectrum, indicating the presence of phosphorus and nitrogen in its char residue. The same peaks also presented in the XPS spectrum of the char residues of the PM/PLA; thus, the phosphorus and nitrogen were likely provided by the PA and melamine components. The presence of phosphorus enhanced the density of the char layer, thereby improving the heat resistance of the composites significantly.

Detailed information about the chemical structure of char residues of PMrG/PLA after CCT can be demonstrated by high-resolution XPS spectra. As displayed in Figure 10a, C 1s of the char residue featured peaks at 284.4 eV (C-C) and 286.5 eV (C=O). For the O 1s spectra (Figure 10b), two peaks were observed, which were attributed to = O in the phosphate or carbonyl groups (531.3 eV) and -O- in the C-O-C or/and C-O-P groups (533.1 eV) [33]. The P 2p peak in Figure 10c appeared around 133.8 eV and was attributed to the P-O-C structure; the N 1s main peak of the char residues at 400 eV was ascribed to the formation of oxidized nitrogen compounds (Figure 10d). The results above show the formation of a cross-linked network in the char residue after the burning of the PMrG/PLA composite.

Based on the above results, the flame retardant mechanism for PMrG-3/PLA is proposed, which is shown in Figure 11. The rGO in PMrG functions as a skeleton, on which melamine and PA are tightly attached through the self-assembling process. As the main framework of PMrG with multilayer structure, rGO provides a barrier effect, which decreases the heat release rate of the composite. The dehydration of the PA and the decomposition of the melamine occurs simultaneously with the burning of the composite, generating non-flammable volatiles, such as H_2_O and NH_3_, which dilute the oxygen concentration on the surface of the composite; thus, the total heat release is significantly decreased when PMrG is used as the flame retardant. In addition, some thermally stable structures, such as P-O-C, are formed by rGO and the decomposition product of PA and melamine, resulting in a cross-linked carbon network in the char residue, which is further densified by the phosphorus element. Thus, for PMrG/PLA composites, a compact char layer with no cracks or cavities is formed during the combustion, which effectively suppresses the release of the heat and smoke. Therefore, rGO, melamine, and PA all play important roles in the high flame retardant performances of PMrG/PLA composites. The inadequacy of the melamine and PA in the PMrG is unfavorable to carbon network formation during combustion, resulting in incompact char residue and a deteriorated heat transfer suppression effect. By contrast, when excess melamine and PA are assembled into PMrG, the amount of rGO is decreased due to the defined PMrG content (10%). The lack of rGO skeleton also influences the formation of cross-linked carbon networks during combustion, resulting in deteriorated flame retardant properties. The most intact and compact char layer only forms when the ratio of rGO, melamine, and PA in PMrG is controlled at a proper value (1:1:5), which endows PMrG/PLA composites with the best flame retardancy.

## 4. Conclusions

This study demonstrates a simple and green synthesis method for a novel flame retardant via the self-assembly of melamine and phytic acid (PA) onto the rGO skeleton, and the composition can be adjusted by changing the ratio of rGO, melamine, and PA. A prepared flame retardant (PMrG) is added into PLA, and it shows the best flame retardant effect on the composite when the ratio of rGO, melamine, and PA is controlled at 1:1:5 (PMrG-3). When 10 wt% PMrG-3 is added, the obtained PMrG-3/PLA composite exhibits greatly reduced pHRR (decreased by 35.3%) and THR (decreased by 21.7%) values, and the TTI is increased from 46 s to 59 s. In addition, the LOI value is increased from 19% to 25%, indicating the excellent flame retardant effect of PMrG-3 on PLA. The improved flame-retardant performance of PMrG-3/PLA composites is attributed to the synergistic effect of the rGO, melamine, and PA components in PMrG, which promote the formation of a compact char layer during the combustion, effectively suppressing the release of the heat and smoke.

## Figures and Tables

**Figure 1 polymers-13-04216-f001:**
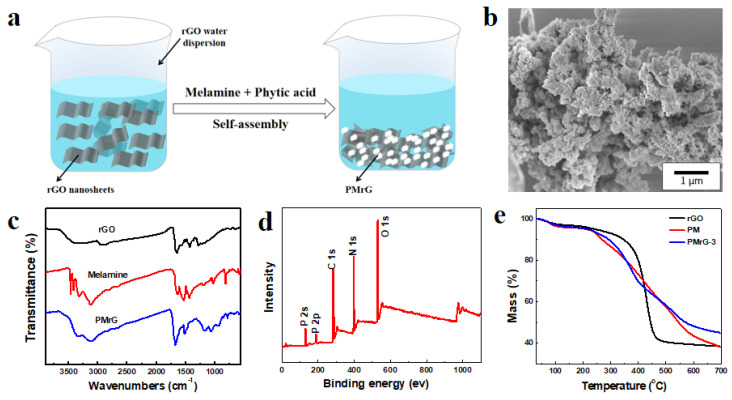
(**a**) Schematic representation for the preparation of PMrG; (**b**) SEM images of PMrG; (**c**) FTIR spectra of rGO, Melamine and PMrG; (**d**) XPS spectra for PMrG; and (**e**) TGA curves of rGO, PM, and PMrG-3.

**Figure 2 polymers-13-04216-f002:**
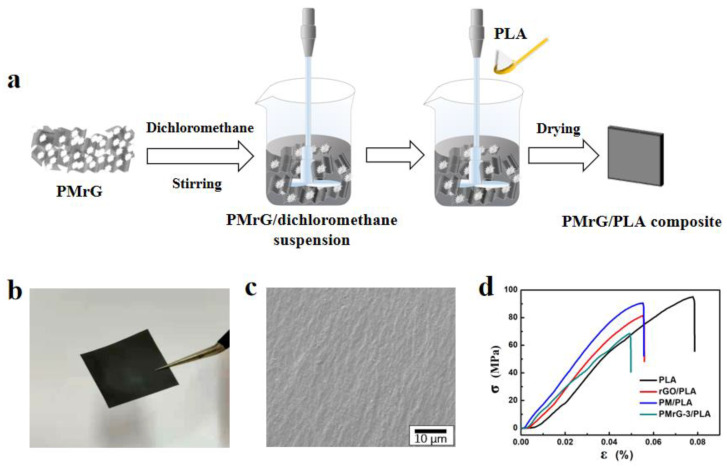
(**a**) Schematic representation for the preparation of PMrG/PLA composite; (**b**) digital photographs of PMrG-3/PLA; (**c**) SEM images of PMrG-3/PLA; and (**d**) stress-strain curves of PLA, rGO/PLA composite, PM/PLA composite, and PMrG-3/PLA composite (the mass fractions of the additives in the composites are all 10%).

**Figure 3 polymers-13-04216-f003:**
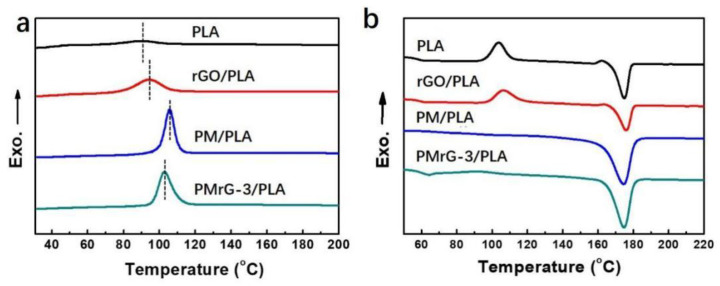
DSC curves of PLA, rGO/PLA composite, PM/PLA composite, and PMrG-3/PLA composite: (**a**) cooling process; (**b**) heating process (the mass fractions of the additives in the composites are all 10%).

**Figure 4 polymers-13-04216-f004:**
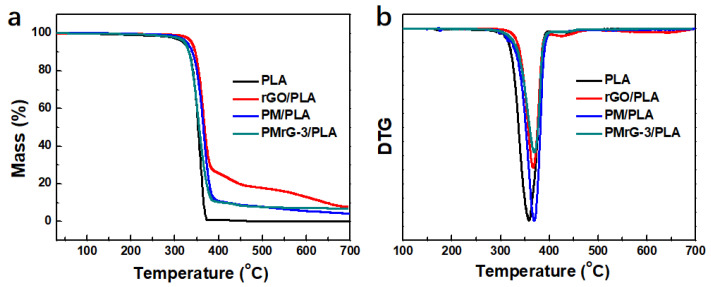
(**a**) TGA and (**b**) DTG curves of PLA, rGO/PLA composite, PM/PLA composite, and PMrG-3/PLA composite (the mass fractions of the additives in the composites are all 10%).

**Figure 5 polymers-13-04216-f005:**
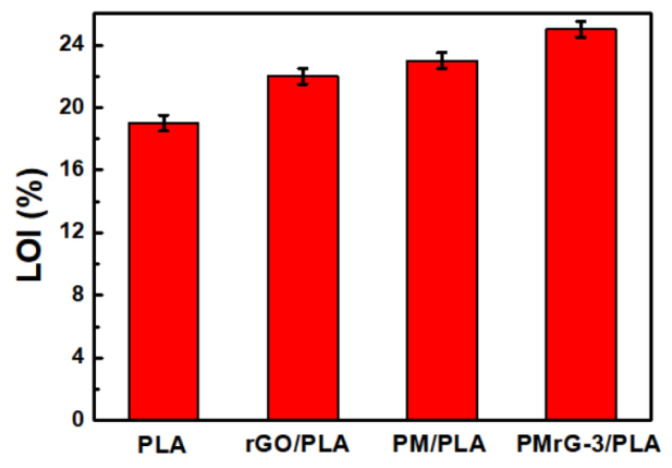
LOI values of PLA, rGO/PLA composite, PM/PLA composite, and PMrG-3/PLA composite (the mass fractions of the additives in the composites are all 10%).

**Figure 6 polymers-13-04216-f006:**
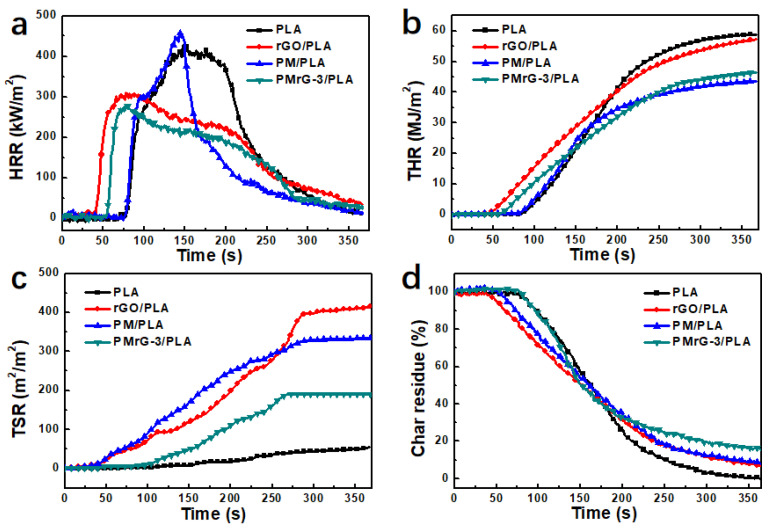
(**a**) Heat release rate, (**b**) total heat release, (**c**) total smoke release, and (**d**) char residue curves of the neat PLA and its composites under an external heat flux of 35 kW/m^2^ (the mass fractions of the additives in the composites are all 10%).

**Figure 7 polymers-13-04216-f007:**
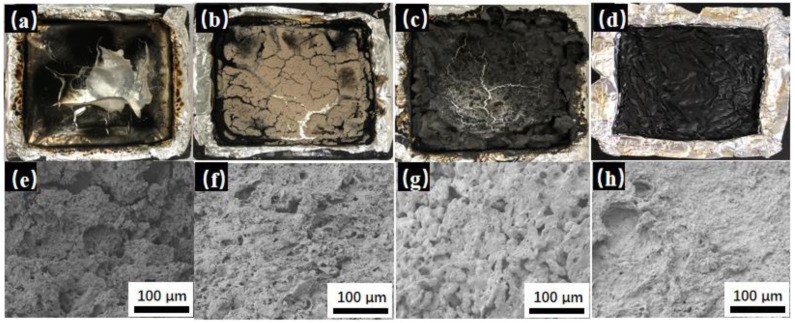
Digital images of char residues for (**a**) PLA, (**b**) rGO/PLA, (**c**) PM/PLA, and (**d**) PMrG-3/PLA after cone calorimeter test; SEM images of char residues for (**e**) PLA, (**f**) rGO/PLA, (**g**) PM/PLA, and (**h**) PMrG-3/PLA after cone calorimeter test.

**Figure 8 polymers-13-04216-f008:**
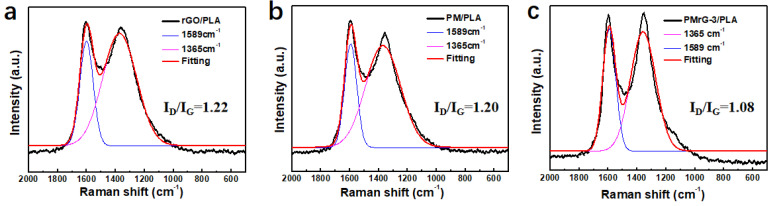
Raman spectra of char residues of (**a**) rGO/PLA, (**b**) PM/PLA, and (**c**) PMrG-3/PLA.

**Figure 9 polymers-13-04216-f009:**
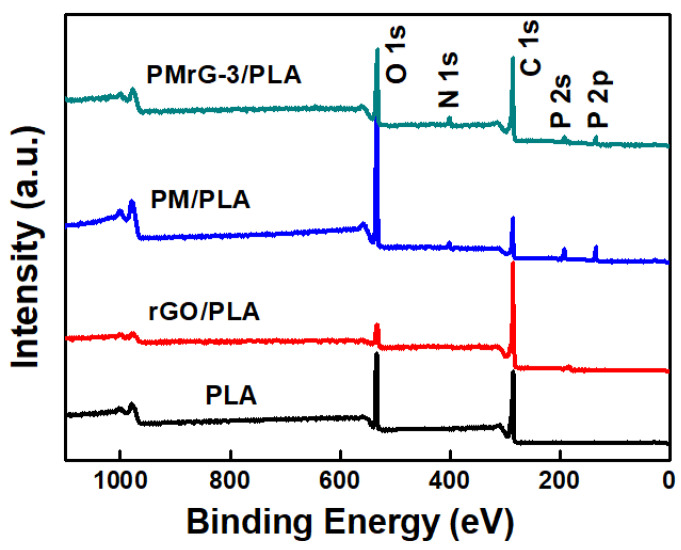
XPS spectra for char residues of the neat PLA and the PLA-based composites.

**Figure 10 polymers-13-04216-f010:**
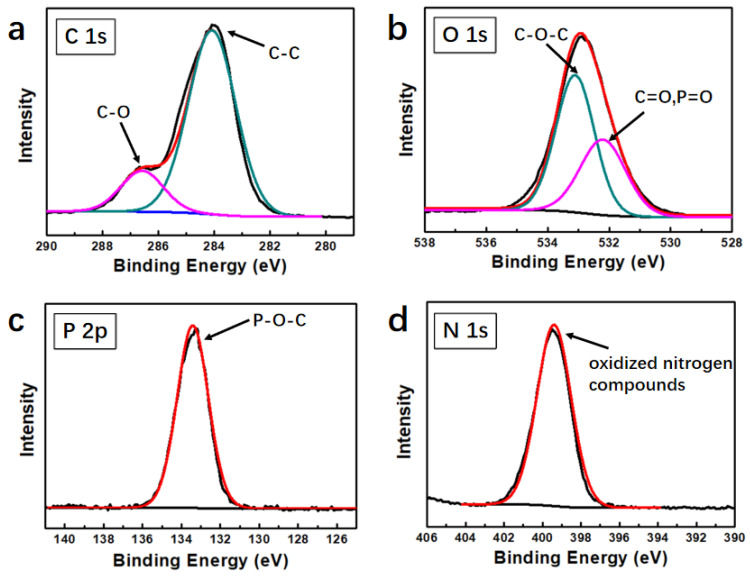
High-resolution XPS spectra for (**a**) C 1s, (**b**) O 1s, (**c**) P 2p, and (**d**) N 1s of the char residue of PMrG/PLA composite.

**Figure 11 polymers-13-04216-f011:**
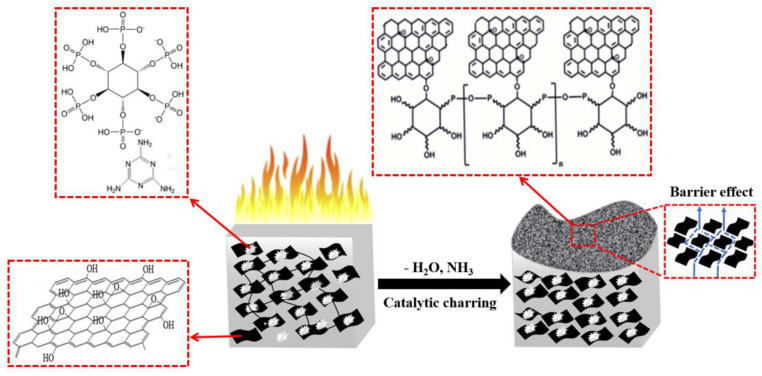
Schematic illustration of flame retardant mechanism.

**Table 1 polymers-13-04216-t001:** Cone calorimeter data of the neat PLA and the PLA-based composites.

Sample	TTI (s)	Phrr (Kw/m^2^)	THR (MJ/m^2^)	TSR (m^2^/m^2^)	Char Residue (%)
PLA	74	426.6	58.8	52.7	0.3
rGO/PLA	43	309.1	58.5	411.7	4.8
PM/PLA	49	456.6	43.2	331.3	6.7
PMrG-3/PLA	59	276.1	46.5	191.5	15.3

## Data Availability

The data presented in this study are available on request from the corresponding author.

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
