# Peer review of "A Novel Self-Assembled Graphene-Based Flame Retardant: Synthesis and Flame Retardant Performance in PLA"

_polymers, 2021, doi:10.3390/polym13234216_

Round 1

Reviewer 1 Report

This manuscript presented an interesting work about the synthesis and evaluation of graphene-based flame retardant in PLA. The work has potential. However, some points listed below need to be improved.

Introduction section: please be more specific about the novelty of this work in the introduction section.

Sectio 2.2: this section must be deeply improved. How the authors prepared PMrG sample? What quantities of reagent were used? What are the conditions used to obtain PMrG sample? Please be more specific.

Section 2.3: What is the ratio of PMrG and dichloromethane used to make the suspension? How the composites were molded?

Section 2.4: How the FTIR analysis was done? KBr pellets or ATR mode?

Author Response

This manuscript presented an interesting work about the synthesis and evaluation of graphene-based flame retardant in PLA. The work has potential. However, some points listed below need to be improved.

Comment 1:Introduction section: please be more specific about the novelty of this work in the introduction section.

Response: Thank you for this reminding. The introduction has been revised carefully, the novelty and superiority of our work has been emphasized in the revised manuscript as following:

In this work, we developed a novel flame retardant for PLA via self-assembly of rGO, melamine, and PA, which is abbreviated as PMrG. The morphology and chemical structure of PMrG was studied. Then, the effects of PMrG on the thermal stability and flame retardancy of PLA were systematically investigated and the mechanism was also discussed. By integrating the phosphorus supplying effect of PA component, the intumescent effect of melamine component, and the flame retardant skeleton constructed by the rGO nanolayers, PMrG exhibits remarkable flame retardant effect on PLA. The PMrG presents the best flame retardant effect on PLA when the ratio of rGO, melamine, and PA is controlled at 1:1:5, which provides the composite with the highest LOI value, the lowest heat release rate (HRR), total heat release (THR), and total smoke release (TSR) values, and the longest time to ignition (TTI) when the amount of the additive keeps at 10 wt%. By using this flame retardant, the composite achieved V-0 grade in UL-94 test. In addition, the pHRR of the composite is decreased by 35%, and the total heat release is decreased by 21%. This work provides a green and facile approach for creating highly effective graphene-based flame retardants for PLA.

Comment 2:Sectio 2.2: this section must be deeply improved. How the authors prepared PMrG sample? What quantities of reagent were used? What are the conditions used to obtain PMrG sample? Please be more specific.

Response: Thank you for your suggestion. The section 2.2 has been deeply improved in the revised manuscript as following:

0.2 g Melamine and 0.9 mL PA was successively added into 100 mL rGO aqueous suspension (1.8 mg mL-1). After the 12 h reaction under mild stirring at the room temperature, the flame retardant system composed of rGO, melamine, and PA can be obtained, which is named PMrG in the following text. Through controlling the ratio of precursors, the PMrG samples with different amount of melamine and PA were synthesized for comparation, and the mass ratio of melamine and PA keeps constant at 1:5 (as shown in Table S1).

In order to illustrate the synergistic effect of rGO, melamine, and PA in PMrG on its flame resistance, we also fabricated a comparative subject by gently stirring 0.2 g melamine and 0.9 mL PA at 100 mL aqueous solution, which is abbreviated as PM.

Comment 3:Section 2.3: What is the ratio of PMrG and dichloromethane used to make the suspension? How the composites were molded?

Response: Thank you for your suggestion. The section 2.3 has been revised as following:

After being washed by using ethanol, 0.6 g prepared PMrG was dispersed in 80 mL dichloromethane to make the PMrG/dichloromethane suspension. Then, 5.4 g PLA was then added into PMrG/dichloromethane suspension under mild stirring to dissolve PLA homogeneously, and the PMrG/PLA/dichloromethane mixture was obtained, in which the mass ratio of PMrG and PLA is controlled at 1:9. Subsequently, the PMrG/PLA/dichloromethane mixture was poured in the Teflon mould and placed at 60 °C under vacuum to remove the dichloromethane solvent until no bubbles emerged, and the PMrG/PLA composite with 10 wt% PMrG was prepared.

Comment 4:Section 2.4: How the FTIR analysis was done? KBr pellets or ATR mode?

Response: The FTIR analysis has been further clarified in the revised manuscript as following:

Fourier transform infrared spectrometry (FTIR) was conducted by a Nicolet 6700 FTIR spectrometer (Thermo Fisher Scientific, Waltham, MA, USA) with an attenuated total reflection (ATR) accessory over the range of 400-4000 cm−1.

Reviewer 2 Report

The flammability of polymers is one of the main problems that limit their use. The addition of flame retardants often affects the mechanical properties and worsens them. When preventing the material from being flammable, it is important to know where it will be used. Each industry has its own fire safety requirements. The first question that arises is where will the flame-retardant PLA be used and what should it meet the minimum criteria. The adopted work methodology is correct. The issue was well-worded. the test methods are well described. The results have been rescheduled. My concern is the value of the oxygen index. In order for the material to be used successfully, its value should be at least 28% or higher. Below this value it is assumed that the material burns easily. OI is the basic criterion for assessing the flammability of plastics. The remaining tests expand our knowledge of these properties. From the researcher's point of view, the work is very interesting and I provide a basis for further work. From the researcher's point of view, the work is very interesting and I provide a basis for further work. However, there is no reference to practical applications and thus criteria. The literature is current and a good sample of the research presented.

Author Response

 The flammability of polymers is one of the main problems that limit their use. The addition of flame retardants often affects the mechanical properties and worsens them. When preventing the material from being flammable, it is important to know where it will be used. Each industry has its own fire safety requirements. The first question that arises is where will the flame-retardant PLA be used and what should it meet the minimum criteria. The adopted work methodology is correct. The issue was well-worded. the test methods are well described. The results have been rescheduled. My concern is the value of the oxygen index. In order for the material to be used successfully, its value should be at least 28% or higher. Below this value it is assumed that the material burns easily. OI is the basic criterion for assessing the flammability of plastics. The remaining tests expand our knowledge of these properties. From the researcher's point of view, the work is very interesting and provides a basis for further work. However, there is no reference to practical applications and thus criteria. The literature is current and a good sample of the research presented.

Response: Thank you for your kind comments and good suggestion. The questions are answered point by point as following:

(1) The application of the flame-retardant PLA has been added in the revised manuscript as following:

However, PLA presents poor flame retardancy, and its limiting oxygen index (LOI) value is only 19. The poor flame retardancy of PLA significantly limited its application, especially in some military fields. The PLA products with improved fire resistance properties are highly desired in the application fields including electronic industry, automotive industry, and aerospace industry [3-5].

[3] Rasal, R.M.; Janorkar, A.V.; Hirt, D.E.; et al. Poly (lactic acid) modifications. Prog Polym Sci 2010, 35, 338-356.

[4] Chalotra, N.; Singh, A.; et al. Rapid Growth and Development of 3D Printing. Int J Eng Sci 2016, 6305-6319.

[5] Chen, Y.; Wang, W.; Qiu, Y.; et al. Terminal group effects of phosphazene-triazine bi-group flame retardant additives in flame retardant polylactic acid composites. Polym Degrad Stab 2017; 140:166-175.

(2)  The most important specific criteria for the flame resistance of the materials is whether it meets the V-0 grade in UL-94 test. In our work, the PLA composite achieved V-0 grade in UL-94 test without melt dripping under the function of 10 wt% PMrG-3. In addition, as another important parameter presenting the flammability of the plastics, the limiting oxygen index (LOI) value of PLA has been improved from 19% to 25% after addition of 10 wt%  PMrG as the flame retardant. The LOI value is supposed to be further increased to above 28% by adding more PMrG into PLA, which will be studied in our future research. The explaination has been added in the revised manuscript.

Round 2

Reviewer 1 Report

After corrections the manuscript reads well. I suggest publication.

Reviewer 2 Report

I have no comments on the revised version, the authors' explanations are satisfactory.